# Age-Dependent Serotype-Associated Case-Fatality Rate in Invasive Pneumococcal Disease in the Autonomous Community of Madrid between 2007 and 2020

**DOI:** 10.3390/microorganisms9112286

**Published:** 2021-11-03

**Authors:** Sara De Miguel, Pello Latasa, José Yuste, Luis García, María Ordobás, Belén Ramos, Marta Pérez, Maira Alejandra Ortiz, Juan Carlos Sanz

**Affiliations:** 1Epidemiology Department, Dirección General de Salud Pública, 28035 Madrid, Spain; sarade.miguel@salud.madrid.org (S.D.M.); plzamalloa@gmail.com (P.L.); luis.garcia@salud.madrid.org (L.G.); maria.ordobas@salud.madrid.org (M.O.); mairaalejandra@gmail.com (M.A.O.); 2Departamento de Epidemiología y Salud Pública, Epidemiología de las Enfermedades Infecciosas, Universidad de Alcalá, 28801 Alcalá de Henares, Madrid, Spain; 3Spanish Pneumococcal Reference Laboratory, Centro Nacional de Microbiología, Instituto de Salud Carlos III, 28220 Madrid, Spain; jyuste@isciii.es; 4CIBER de Enfermedades Respiratorias (CIBERES), 28029 Madrid, Spain; 5Unidad de Microbiología Clínica, Laboratorio Regional de Salud Pública de la Comunidad de Madrid, Dirección General de Salud Pública, 28055 Madrid, Spain; belen.ramos@salud.madrid.org (B.R.); mpabeledo@salud.madrid.org (M.P.); 6CIBER de Epidemiología y Salud Pública (CIBERESP), 28029 Madrid, Spain

**Keywords:** *Streptococcus pneumoniae*, invasive pneumococcal disease, serotype, vaccine, fatality

## Abstract

The aim of this study was to investigate the serotype-associated fatality rate in cases of invasive pneumococcal disease (IPD) in the Spanish region of Madrid between 2007 and 2020. Serotyping was performed by Pneumotest Latex and the Quellung reaction using commercial antisera. Case-fatality rate was estimated as the ratio between the number of deaths at hospital discharge and the number of cases attributable to each serotype. To evaluate the association measures, the odds ratios with a 95% confidence interval were calculated. Twenty five pneumococcal serotypes were associated to mortality and comprised 87.8% of the total number of isolates characterized. Serotypes 8, 3, 19A, 1, 7F, 22F, 12F, and 11A were the most prevalent (≥3% each). Serotypes 31, 11A, and 19F were significantly associated to high case-fatality rates (>20% each). The lower significantly associated case-fatality rate (<10% each) was found in serotypes 5, 1, 12B, 7F, 12F, 8, 33, and 10A. The serotypes with higher mortality levels (≥0.04 per 100,000 population) were 11A (fatality 24.0%), 3 (fatality 18.7%), 19A (fatality 12.5%), and 8 (fatality 7.2%). Serotype 3 was worrisome because it is associated with important fatality levels combined with very high incidence and mortality rates. Serotype 11A also showed a high fatality with marked incidence and mortality levels. Some few frequent serotypes as 31, 19F, and 15A despite its high fatality had low levels of mortality. By contrast other serotypes as 8 showing low fatality had high mortality ranges because it shows a wide extended distribution. Finally, common serotypes, such as 1 and 5, presented small mortality length, due to their low case-fatality rates.

## 1. Introduction

In 2001, the seven-valent pneumococcal conjugate vaccine (PCV7) comprising serotypes 4, 6B, 9V, 14, 18C, 19F, and 23F was approved for pediatric use in Europe. From 2006–2008, it was introduced into the national pediatric immunization programs for many European countries. From 2009–2011, it was replaced by new available conjugate vaccines against 10 serotypes (PCV10, adding to PCV7 the serotypes 1, 5, 7F) and 13 serotypes (PCV13, adding to PCV10 serotypes 3, 6A, and 19A). The serotype diversity increased and varied by age. Among serotypes that have become more frequent in countries which employ PCV13 are 24F, 22F, 8, and 15A, while 19A and 3 are used in countries that use PCV10 [1]. With the introduction of pneumococcal conjugate vaccines, the interest to monitor the distribution of vaccine covered serotypes and to detect the emergence of non-vaccine serotypes has become a paramount deal. The relationship between pneumococcal serotypes and antibiotic resistance now it is well recognized [2]. After PCV7 use, most countries reported a decrease in antibiotic resistance [2]. However, universal use of PCV7 led to the increase in certain non-vaccine serotypes, such as 19A, which was associated with multidrug resistance, therefore contributing to recover resistance rates for pneumococcus. In this sense, in Massachusetts between 2007 and 2009, PCV7 serotypes accounted for 7% whereas PCV13 serotypes accounted for 77% of all cases [3]. Invasive pneumococcal disease (IPD) increased due to serotypes 19A and 7F, while 19A and 7F were isolated in 41% and 20% of all IPD cases in the same period, respectively. Serotype 19A also comprised a majority of the penicillin non susceptible isolates [3]. The introduction of PCV13 further reduced the incidence of IPD and decreased antibiotic resistance on pneumococci [4]. Although, overall incidence of IPD has declined after the use of pneumococcal conjugate vaccines, the disease features changed shifting towards more severe clinical forms [5]. The IPD is a mandatory reporting disease included from 2007 in the surveillance system of the Autonomous Community of Madrid [6] (BOCM 2007). The serotype knowledge generated in recent years, including clinical information, makes it now possible to not only investigate epidemiological (e.g., vaccine impact) and microbiological (e.g., antimicrobial susceptibility trends) factors but also clinical strain-specific risk for severe or fatal outcomes too [7]. The aim of this study was to investigate the serotype-associated fatality rate in cases of IPD in the Autonomous Community of Madrid between 2007 and 2020.

## 2. Materials and Methods

The analysis focuses on IPD cases registered between 2007 and 2020 in the Notifiable Diseases Surveillance System covering the total resident population of the Autonomous Community of Madrid. Doctors and microbiologists from all public and private hospitals, clinics, and laboratories report the disease to the Epidemiological Surveillance Network, which provides clinical and microbiological information. *Streptococcus pneumoniae* strains isolated from normally sterile clinical samples in public and private hospitals of Madrid were identified by standard methods. Serotype was centralized in the Madrid Regional Public Health Laboratory, and was performed by Pneumotest Latex and Quellung reaction using commercial antisera (Statens Serum Institute, Copenhagen, Denmark). Case-based information was collected, and the report and the medical history was consulted using a standardized form that included sociodemographic information, clinical presentation, and patient evolution during hospitalization and existence of risk factors. Fatality was defined as an IPD-related death that are fatal within 28 days of onset. The information is validated, completed, and recorded by the epidemiologists of Madrid in the notifiable diseases surveillance system and exported for analysis. The variables included in the analysis were, sex, clinical presentation (pneumonia, bacteremia, meningitis sepsis and a cluster of other infections of usually sterile sites as [empyema, peritonitis and arthritis]), antecedents of risk factors (immunodeficiency, splenectomy, surgery, cerebrospinal fluid fistula, traumatic brain injury, respiratory pathology, cardiac, hepatic, renal or other chronic pathologies), and year on notification. A total of 7851 IPD cases with the start of symptoms between 2007 and 2020 were notified. A total of 921 IPD cases were excluded by unknown clinical outcome and 917 because the strain was not available for serotyping. Hence, 6013 cases were included in the study. The average incidence and average mortality rates were calculated, respectively, by dividing the number of cases or number of deaths caused by each of the serotypes among the sum of the populations monitored each year during the 2007–2020 study period (90537248 inhabitants) [8]. The specific fatality rate was calculated as the number of deaths among the number of total cases by each serotype. The 25 more frequent serotypes were selected based in their frequency causing IPD cases and deaths by each serotype. The crude and adjusted odds ratios (ORc and ORa) with its 95% confidence interval (CI95%) were calculated by means of logistic regression models.

## 3. Results

The main characteristics of cases are shown in Table 1.

The age showed a bimodal distribution, affecting mainly to the age groups of 0–14 years (18.21%), 50–64 years (19.72%), and older than 65 years (40.22%). Fatality rate significantly increased with age, from 2.16% in younger to 28.88% in older people (85 years or more) (OR: 28.86; CI95%: 17.2 to 48.5). The rate was not significantly higher in males than in females (53.1% vs. 42.9%). The most frequent clinical presentation was pneumonia (53.9%) followed by bacteremia without focus (16.2%). The fatality rate varied according to the clinical presentation being significantly higher in sepsis (OR: 5.47; CI95%: 4.5 to 6.6) and meningitis (OR: 1.82; CI95%: 1.4 to 2.4). Furthermore, 57.6% of cases had at least one risk factor (immunodeficiency; splenectomy; surgery; cerebrospinal fluid fistula; traumatic brain injury; and respiratory, cardiac, hepatic, renal, or other chronic pathologies). The fatality rate was significantly higher in patients with risk factors (OR: 3.28; CI 95%: 2.7 to 4.0). The number of cases by year ranged from 297 in 2013 to 574 in 2009. In 2020, the number of reported cases of IPD was significantly lower that the annual trend. In general, no statistical differences in fatality rates throughout the different years. The incidence, mortality and serotype-specific fatality rates are shown in Table 2.

Twenty-five serotypes comprised 87.8% of the isolates. The more frequent serotypes were 8 (16.5%; incidence 1.09 per 100,000 population), 3 (10.1%; incidence 0.67 per 100,000), 19A (7.6%; incidence 0.5 per 100,000), 1 (6.9%; incidence 0.46 per 100,000), 7F (4.8%; 0.31 incidence per 100,000), 22F (4.0%; incidence 0.26 per 100,000), 12F (3.2%; 0.21 incidence per 100,000), and 11A (3.0%; incidence 0.20 per 100,000). The specific fatality rate varied among serotypes (Figure 1).

The serotypes with higher fatality rates were 31 (29.5%), 11A (24.0%), 19F (21.8%), 15A (19.1%), and 35B (18.7%), which is statistically significant for serotypes 31, 11A, 19F, and 15A (*p* < 0.05). The lower fatality rates were found in cases by serotypes 5 (0.9%), 1 (%1.0), 12B (5.4%), 7F (5.6%), and 12F (5.6%). The serotypes with the highest mortality rates were 3 (0.13 per 100,000), 8 (0.08 per 100,000), 19A (0.06 per 100,000), and 11A (0.05 per 100,000). Table 3 shows the cases of deaths by IPD in the last 3 years by serotype. Serotypes 3 and 8 have caused more deaths with 13.8% and 6.8% fatality rate. The serotypes with the higher fatality rates were 11D (100%), 24A (50%), and 23F (40%); however, they had a low mortality.

## 4. Discussion

In this study, a long-term period of 13 years was monitored for IPD in Madrid to analyze mortality by serotypes [9]. This fact allows us to collect a number of strains of different serotypes and raise statistical power. However, distribution, incidence, and mortality of serotypes change along the time and, hence, an important limitation of this study is that we can obtain a temporal “pooled static picture” of the entire 2007–2020 interlude. Other limitations of this study rests that the evolution of serotypes, their distribution, as well as antibiotic resistance, or changes in virulence factors were not taken into account.

In 2020, there was a strong decrease in the number of IPD due to the COVID-19 pandemic. The decrease could be attributed to the lower notification rates due to the overload of public health resources and the lower transmission due to the lockdown, the social distancing measures, or the general use of face masks.

Another weakness of this study is the identification of deceased cases, which only included in-hospital patients [10,11]. This aspect may have somehow misjudged the real case-fatality rate. Because of the lack of some relevant information (non-identified serotypes or lack of clinical evolution outcome), the incidence and serotype-specific mortality were calculated according to the number of cases included in the study and not to the real number of notified cases. This fact constitutes an important bias that underestimates the real incidence and mortality by serotype. However, this assessment allows us to evaluate the burden of fatality rate in the mortality in connection with the incidence of every serotype. Our results show that twenty-five serotypes comprised more than 85% of the isolates. This is consistent with recent estimates at the national level of IPD cases in Spain [12].

The highest fatality rates were detected in cases by serotypes 31, 11A, 19F, 15A, 35B, 6C, 3, 23A, 9V, and 9N. In other recent series, the highest fatality rates were for cases by serotypes 31, 11A, 19F, 6A, and 3 [13]. Some less frequent serotypes, such as 31, 19F, and 15A, despite its high fatality rate (>20%), had low levels of mortality (<0.03 per 100,000 population). By contrast, other serotypes showing moderate (serotype 19A) or low (serotype 8) fatality rates had high mortality rates because of their wide extended distribution. Finally, common serotypes, such as 1, 7F, and 22F, presented low mortality rates (0.00 per 100,000, 0.02 per 100,000 and 0.04 per 100,000), due to their low fatality rates (0.96%, 5.6%, and 15.1%, respectively). Several intrinsic aspects related to the bacterium itself might contribute to the increase in fatality rates, such as harboring antibiotic resistance or the expression of certain virulence factors. In this sense, isolates of serotype 11A with high fatality rates in our study increased in the last years, mainly due to strains associated with high levels of β-lactam resistance [14]. The reason is the emergence of specific clones among serotype 11A with a higher ability to avoid complement-mediated immunity and increased biofilm formation capacity [14].

Changes in the incidence of each serotype are undoubtedly influenced by the introduction of vaccination [9]. During the period 2007–2020, the immunization program in Madrid underwent many changes which affect the pediatric population but also to the adult vaccination policy. The PCV7 was introduced in November 2006 into the public funding immunization pediatric program of Autonomous Community of Madrid. In June 2010, the PCV13 replaced PCV7, but it was excluded in July from the regional immunization program because of changes in vaccination policies. From this time, the vaccine was available at private markets. Finally, PCV13 was reintroduced in December 2014 in the pediatric vaccination program [9]. In Europe, after the introduction of PCV13 alone or PCV10 followed by PCV13, the incidence of IPD decreased due to a decline in the incidence of vaccine serotypes. By contrast, IPD caused by non-PCV13 serotypes increased, suggesting the phenomena of serotype replacement [12,15]. Hence, while PCV13-covered serotypes continued to decrease in all age groups as a consequence of herd immunity, it is expected that mortality by these serotypes will diminish in the near future. This is important for the previously considered intermediate case-fatality rate serotype 19A [16]. In Spain, its incidence has decreased in children and adults [12], although its fatality rate represents 12.5% throughout the period studied and increases to 14.3% in the last 3 years.

The importance of serotype 3 is clear because it is one of the most prevalent PCV13 serotypes and is associated with high mortality and case-fatality rates, mostly in old infected patients. This serotype has been classified among the most fatal [11,13,16,17] and is associated with increased risk of death [18]. Although, after a continuous vaccine use, the mortality associated to this serotype will descend too, the slow reduction in its incidence would delay this fall [19,20]. In our study, the fatality rate of serotype 3 represents 18.7%, and decreases to 13.8% in last 3 years; however, serotyping was limited to culture-isolated strains. In this sense, serotype 3 is only detectable by PCR in some IPD episodes [21] and, therefore, the contribution of this serotype to fatality rate could be underestimated.

Some serotypes, such as 1 and 7F, associated to high invasive disease potential, are carried for short periods and they seem to affect to relatively healthy adults [16,22,23]. Low case-fatality rate of serotype 1 has been described previously [16], with near elimination of IPD in all ages [24], showing a decreased risk of dying [18]. Although serotype 7F had been described in some studies as quite lethal in children [7], it is usually considered as a low case-fatality rate [16] and is associated with a small risk of death [18]. Both show a low case-fatality rate in the last three years. For this reason, the impact of the PCV13 vaccination in mortality by both of these serotypes may be less important.

Serotype 5 has also a very low case-fatality rate [16] and is linked to community outbreaks [25]. This serotype started to diminish in the Autonomous Community of Madrid before the introduction of PCV13 [26], and has practically vanished since 2011 [27].

In these series, the most frequent was the non-PVC13 serotype 8. Although this serotype showed an increased odds of death in meningitis cases [28], in general, it has been considered of intermediate case-fatality by other authors [16] and of low death risk [18]. Nonetheless, in order of mortality with a fatality rate around 6.8% in last three years, it was the second (after serotype 3), due to its elevated incidence in these series. It is a rare serotype in children but is frequently associated in adults [4,29].

The non-PCV13 serotypes 22F, 11A, and 24F have been formerly classified as of intermediate case-fatality rate [16]. In these series, serotype 11A showed a significantly high case-fatality rate. This fact, and the associated high incidence and mortality rate of this serotype, constitute a matter of concern. Furthermore, 22F is rising and increasing its proportion over total serotypes in the pediatric population [30]; its fatality rate over the last three years represents 13.3%.

It is interesting to note that very rare serotypes (every those under 3% on the total) may reach a significant rate of case-fatality rate. Invasive capacity has been computed dividing the incidence of IPD due to a specific serotype by the carriage prevalence of this serotype in persons of the same age [31]. Serotypes with increased relative risk of fatal outcome had usually high carriage prevalence and low invasiveness [18]. The mortality is dependent of serotype frequency and serotype incidence.

The case-fatality rate is an independent factor of incidence that may be related to interactions between the bacteria and the host. Despite this, differences in incidence and case-fatality rate might, in part, also reflect differences of serotypes and clones with different virulence potential [32]. Case-fatality rate varies obviously according to the patient underlying conditions, increasing with age, immuno-compromise conditions, and co-morbidities [32,33,34,35,36]. The highest risk of death was among patients with solid tumors, followed by patients with hemodialysis, cardiovascular disease, and hematological malignancy [32]. Mortality has been associated in older adults with age >85 years and >1 high-risk conditions (high risk, including chronic renal failure, HIV, immunodeficiency (medically induced or innate), asplenia, hematological or metastatic malignancies, CSF leak, and prior neurosurgery) [37]. In this study, we considered the variable of “risk factors” gathering different factors (immunodeficiency; splenectomy; surgery; cerebrospinal fluid fistula; traumatic brain injury; or respiratory, cardiac, hepatic, renal, or other chronic pathologies). Hence, one important drawback of this paper concerns the lack of consideration for specific underlying diseases. As observed in other series, case-fatality rate was highest among patients with septicemia [32]. Meningitis is also commonly associated to increased case-fatality rates [32,38]. In children, the presentation as meningitis had six-fold higher case-fatality rates compared with children with isolated bacteremia [35]. In this study, the fatality rate was significantly higher in meningitis than in other invasive clinical forms. This finding may be related to the age on this clinical presentation, usually more frequent in children than older adults (data not show). Among studies in adults, some serotypes, such as serotype 3, have been associated with an elevated risk of necrotizing pneumonia and septic shock, although no data of antimicrobial susceptibility were regarded [17,39]. In this sense, non-susceptibility to penicillin has also been significantly associated with death [40]; however, this fact would be a consequence linked to the expansion of the multi-resistant serotype 19A [41]. It has been proposed that other serotypes, such as 3 and 8, are more prone to affect more fragile or older individuals [16]. Serotype 8 has been shown to be highly invasive [42,43], while serotype 3 has been considered highly invasive by some authors [23,31,43] and lower [44] or variable by others [42].

In conclusion, considering that the risk of death is a stable serotype-associated property [18], investigation of case-fatality serotype associate rates is of vital importance for epidemiological surveillance [40]. The identification of more lethal serotypes or those with a high propensity to affect risk groups can be important for the composition of future conjugate vaccines [33]. Serotypes 22F and 33F are associated to high invasive capacity [30,31] and are candidates to the new 15-valent pneumococcal conjugate vaccine (PCV15) [45].

The recently patented 20-valent pneumococcal conjugate vaccine (PCV20) would add to the PCV15 the five new serotypes (8, 10A, 11A, 12F, and 15B) [46]. The inclusion of serotype 8 (low to moderate case-fatality rate but high incidence and mortality) and, to a smaller extent, serotype 11A (high case-fatality rate but lower incidence and mortality) could be especially relevant for adults [29]. These additional serotypes are crucial in our region from an epidemiological perspective and they would help to control the impact of IPD in the next few years. Besides, serotype-specific case-fatality rate seems to be a key complement to other epidemiological aspects that might add very practical prognostic information that could be very useful for clinicians attending IPD cases.

## Figures and Tables

**Figure 1 microorganisms-09-02286-f001:**
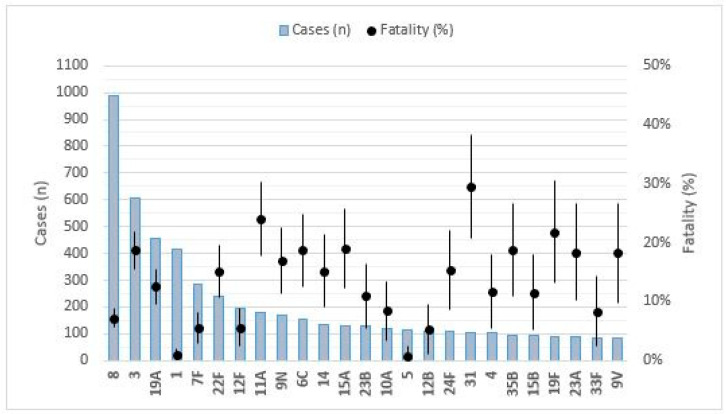
The number of IPD cases in relation to the fatality rate of every serotype. Grey bars indicate the number of IPD cases by each serotype. Black dots indicate the proportion of fatality by each serotype.

**Table 1 microorganisms-09-02286-t001:** Multivariable logistic regression to assess independent risk factors for death. Description of the studied population. Invasive Pneumococcal Disease. Mandatory Reportable Diseases. Autonomous Community of Madrid. Years 2007 to 2020.

	Cases		Deaths				
	N	%	N	%	CI (95%)	OR	CI (95%)
**Age group**							
0–14	1095	18.21	16	2.16	(0.8–2.2)	1	
15–34	329	5.47	8	1.08	(0.8–4.1)	1.68	(0.7–4.0)
35–50	984	16.36	70	9.45	(5.5–8.7)	5.16	(3.0–9.0)
50–64	1186	19.72	148	19.97	(10.6–14.4)	9.62	(5.7–16.2)
65–74	815	13.55	103	13.90	(10.4–14.9)	9.76	(5.7–16.7)
75–84	890	14.80	182	24.56	(17.8–23.1)	17.34	(10.3–29.2)
≥85	714	11.87	214	28.88	(26.6–33.3)	28.86	(17.2–48.5)
**Sex**							
Female	2577	42.86	303	40.89	(10.5–13.0)	1	
Male	3436	57.14	438	59.11	(11.6–13.9)	1.10	(0.9–1.3)
**Clinical presentation**							
Pneumonia	3239	53.87	256	34.55	(7.0–8.8)	1	
Bacteraemia	975	16.21	97	13.09	(8.1–11.8)	1.29	(1.0–1.6)
Meningitis	511	8.50	69	9.31	(10.5–16.5)	1.82	(1.4–2.4)
Other *	433	7.20	46	6.21	(7.7–13.5)	1.39	(1.0–1.9)
Sepsis	855	14.22	273	36.84	(28.8–35.1)	5.47	(4.5–6.6)
**Risk Factors ****							
No	2546	42.34	150	20.24	(5.0–6.8)	1	
Yes	3467	57.66	591	79.76	(15.8–18.3)	3.28	(2.7–4.0)
**Year**							
2007	446	7.42	59	7.96	(10.1–16.4)	1	
2008	533	8.86	56	7.56	(7.9–13.1)	0.77	(0.5–1.1)
2009	574	9.55	48	6.48	(6.1–10.6)	0.60	(0.4–0.9)
2010	410	6.82	37	4.99	(6.2–11.8)	0.65	(0.4–1.0)
2011	422	7.02	54	7.29	(9.6–16.0)	0.96	(0.6–1.4)
2012	330	5.49	57	7.69	(13.2–21.4)	1.37	(0.9–2.0)
2013	297	4.94	49	6.61	(12.3–20.7)	1.30	(0.9–2.0)
2014	373	6.20	54	7.29	(10.9–18.1)	1.11	(0.7–1.7)
2015	419	6.97	62	8.37	(11.4–18.2)	1.14	(0.8–1.7)
2016	447	7.43	64	8.64	(11.1–17.6)	1.10	(0.7–1.6)
2017	586	9.75	67	9.04	(8.9–14.0)	0.85	(0.6–1.2)
2018	515	8.56	61	8.23	(9.1–14.6)	0.88	(0.6–1.3)
2019	554	9.21	50	6.75	(6.6–11.4)	0.65	(0.4–1.0)
2020	107	1.78	23	3.10	(13.7–29.3)	1.80	(1.1–3.1)

* Other clinical presentation: empyema, peritonitis, arthritis, other infections of usually sterile sites. ** Risk factors: immunodeficiency, splenectomy, surgery, cerebrospinal fluid fistula, traumatic brain injury, respiratory pathology, cardiac, hepatic, renal or other chronic pathologies.

**Table 2 microorganisms-09-02286-t002:** Description of the studied population and multivariate logistic regression analysis of invasive pneumococcal disease-related mortality, according to description variables and relevant risk factors. Autonomous Community of Madrid. Years 2007 to 2020.

SEROTYPE	CASES (N)	INCIDENCE +	DEATHS (N)	MORTALITY +	FATALITY (%)	ORC (CI 95%)	P-CRUDE	ORA (CI 95%)	P-ADJUSTED
**8**	990	1.09	71	0.08	7.17	0.50 (0.4–0.6)	0.00 *	0.50 (0.4–0.6)	0.00 *
**3**	605	0.67	113	0.12	18.68	1.75 (1.4–2.2)	0.00 *	1.35 (1.1–1.7)	0.01 *
**19A**	455	0.50	57	0.06	12.53	1.02 (0.8–1.4)	0.89	1.20 (0.9–1.6)	0.25
**1**	417	0.46	4	0.00	0.96	0.06 (0.0–0.2)	0.00 *	0.16 (0.1–0.4)	0.00 *
**7F**	286	0.32	16	0.02	5.59	0.41 (0.2–0.7)	0.00 *	0.61 (0.4–1.0)	0.07
**22F**	238	0.26	36	0.04	15.13	1.28 (0.9–1.8)	0.18	1.08 (0.7–1.6)	0.68
**12F**	195	0.22	11	0.01	5.64	0.42 (0.2–0.8)	0.01 *	0.49 (0.3–0.9)	0.02 *
**11A**	179	0.20	43	0.05	24.02	2.33 (1.6–3.3)	0.00 *	1.93 (1.3–2.8)	0.00 *
**9N**	172	0.19	29	0.03	16.86	1.46 (1.0–2.2)	0.07	1.28 (0.8–2.0)	0.25
**6C**	155	0.17	29	0.03	18.71	1.66 (1.1–2.5)	0.02 *	1.12 (0.7–1.7)	0.60
**14**	132	0.15	20	0.02	15.15	1.28 (0.8–2.1)	0.32	1.01 (0.6–1.7)	0.98
**15A**	131	0.14	25	0.03	19.08	1.70 (1.1–2.6)	0.02 *	1.43 (0.9–2.3)	0.13
**23B**	128	0.14	14	0.02	10.94	0.87 (0.5–1.5)	0.63	1.05 (0.6–1.9)	0.87
**10A**	118	0.13	10	0.01	8.47	0.65 (0.3–1.3)	0.20	0.71 (0.4–1.4)	0.32
**5**	116	0.13	1	0.00	0.86	0.06 (0.0–0.4)	0.01 *	0.16 (0.0–1.2)	0.07
**12B**	111	0.12	6	0.01	5.41	0.40 (0.2–0.9)	0.03 *	0.44 (0.2–1.0)	0.06
**24F**	111	0.12	17	0.02	15.32	1.29 (0.8–2.2)	0.33	1.47 (0.8–2.6)	0.17
**31**	105	0.12	31	0.03	29.52	3.07 (2.0–4.7)	0.00 *	1.81 (1.2–2.8)	0.01 *
**4**	103	0.11	12	0.01	11.65	0.94 (0.5–1.7)	0.83	1.12 (0.6–2.1)	0.72
**35B**	96	0.11	18	0.02	18.75	1.66 (1.0–2.8)	0.06	1.28 (0.7–2.2)	0.38
**15B**	95	0.10	11	0.01	11.58	0.93 (0.5–1.8)	0.82	1.08 (0.6–2.1)	0.83
**19F**	87	0.10	19	0.02	21.84	2.01 (1.2–3.4)	0.01 *	2.17 (1.3–3.8)	0.01 *
**23A**	87	0.10	16	0.02	18.39	1.62 (0.9–2.8)	0.09	1.30 (0.7–2.3)	0.36
**33**	84	0.09	7	0.01	8.33	0.64 (0.3–1.4)	0.27	0.70 (0.3–1.6)	0.39
**9V**	82	0.09	15	0.02	18.29	1.61 (0.9–2.8)	0.10	1.05 (0.6–1.9)	0.88

^+^ Per 100,000 population. * *p* < 0.05.

**Table 3 microorganisms-09-02286-t003:** Mortality and fatality rates by serotype in IPD deaths over the last 3 years. Autonomous Community of Madrid. Years 2019 to 2020.

Serotype	Deaths	Cases	Mortality *	Fatality (%)
**11D**	1	1	0.01	100.00
**24A**	1	2	0.01	50.00
**23F**	2	5	0.03	40.00
**6C**	11	36	0.16	30.56
**37**	1	4	0.01	25.00
**23A**	5	22	0.07	22.73
**24F**	5	22	0.07	22.73
**11A**	8	38	0.12	21.05
**20**	3	15	0.04	20.00
**31**	4	20	0.06	20.00
**11B**	1	5	0.01	20.00
**15B**	3	15	0.04	20.00
**19F**	3	15	0.04	20.00
**35F**	3	15	0.04	20.00
**13**	1	6	0.01	16.67
**12A**	1	6	0.01	16.67
**15A**	6	37	0.09	16.22
**19A**	4	28	0.06	14.29
**25A**	2	14	0.03	14.29
**35B**	2	14	0.03	14.29
**9V**	2	14	0.03	14.29
**3**	18	130	0.27	13.85
**22F**	6	45	0.09	13.33
**9N**	5	41	0.07	12.20
**16F**	3	28	0.04	10.71
**17F**	1	10	0.01	10.00
**24B**	1	11	0.01	9.09
**8**	23	340	0.34	6.76
**14**	1	16	0.01	6.25
**4**	1	17	0.01	5.88
**12F**	1	20	0.01	5.00
**12B**	2	47	0.03	4.26
**33F**	1	25	0.01	4.00
**10A**	1	29	0.01	3.45
**1**	0	1	0.00	0.00
**21**	0	5	0.00	0.00
**15C**	0	10	0.00	0.00
**15F**	0	1	0.00	0.00
**18C**	0	4	0.00	0.00
**18F**	0	1	0.00	0.00
**23B**	0	24	0.00	0.00
**35A**	0	2	0.00	0.00
**6A**	0	2	0.00	0.00
**6B**	0	1	0.00	0.00
**7F**	0	7	0.00	0.00
**9A**	0	1	0.00	0.00

* 100,000.

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
