# Peer review of "Age-Dependent Serotype-Associated Case-Fatality Rate in Invasive Pneumococcal Disease in the Autonomous Community of Madrid between 2007 and 2020"

_microorganisms, 2021, doi:10.3390/microorganisms9112286_

Round 1

Reviewer 1 Report

Title: title should contain information also regarding the study design. please add.

Abstract: according to the authors' guidelines abstract should be unstructured. Please revise.

Methods:

please add who and how Case-based information was collected. Now, it is only clear the source of this information. Please add.

Describe the setting and locations

Give the eligibility criteria, and the sources and methods of selection of participants

Please clearly define all outcomes, exposures, predictors, potential confounders, and effect modifiers. Now only incidence, osrtality and fatality rate were defined. Please add all the other variables used.

Describe any efforts to address potential sources of bias

Results: are clearly presented. Please improve the quality of table

Discussion:

The discussion contains almost all the important information needed. However, the structure is a bit difficult to follow. Usually, the discussion is based on the following elements:

  1. Summary of the main results obtained 
  2. internal consistency and comparability of the data obtained
  3. followed by a general interpretation of the results in the context of other evidence
  4. limitations of the study performed
  5. implications of the results for practice, policy, and future research

In this paper even if most of these elements are there, they did not follow the appropriate structure and reader have to grasp into the text in order to found all of them. Please restructure the discussion. Moreover, in my view this study is highly important in terms of public health, but a very few elements and implications are discussed by authors. please add. this is an added value of your work.

Conclusions are now missing. Please add 

Author Response

Response to Reviewer 1 Comments

Point 1: Title: title should contain information also regarding the study design. please add.

We thank the reviewer for the suggestion. We have modified the title as follows:

Age-dependent serotype-associated case fatality rate in invasive pneumococcal disease in the Autonomous Community of Madrid between 2007 and 2020.

Point 2: Abstract: according to the authors' guidelines abstract should be unstructured. Please revise.

Response 2: We thank the reviewer for the reminder. We have made the changes to adjust to the author’s guidelines abstract.

Point 3: please add who and how Case-based information was collected. Now, it is only clear the source of this information. Please add.

Response 3: We thank the reviewer for the suggestion. IPD is a mandatory reporting disease and doctors and microbiologists from all public and private hospitals, clinics and laboratories of the Community of Madrid report the disease to the epidemiological surveillance network providing clinical and microbiological information (please, see “introduction”). The information is validated, completed and recorded by the epidemiologists of Madrid in the Notifiable Diseases Surveillance System that cover the total resident population of the Autonomous Community of Madrid (please, see “Martials and Methods). This information is accessible for epidemiologists of Madrid for analysis and we have exported this information to carry out this study.

We have made the changes in the manuscript including this information.

Point 4: Describe the setting and locations

Response 4: Please see the Response 3

Point 5: Give the eligibility criteria, and the sources and methods of selection of participants

Response 5: We included all reported cases of IPD from the Community of Madrid registered between 2007 and 2020. The database was extracted from data recorded in the Notifiable Diseases Surveillance System for analysis. In “Material and Methods” and in “Discussion” as a limitation, we explain that some cases were excluded for the analysis by unknown clinical outcome and because the strain was not available for serotyping because it could change the outcome of the mortality rate and the real ratio of each serotype.

Point 6: Please clearly define all outcomes, exposures, predictors, potential confounders, and effect modifiers. Now only incidence, mortality and fatality rate were defined. Please add all the other variables used.

Response 6: All outcomes that we have measured in this study are defined when they are cited for first time, mainly in the material and methods section. The variables contemplated are: age, sex, clinical presentation (pneumonia, bacteraemia, meningitis sepsis and a cluster of other infections of usually sterile sites as [empyema, peritonitis and arthritis]), antecedents of risk factors (immunodeficiency, splenectomy, surgery, cerebrospinal fluid fistula, traumatic brain injury, respiratory pathology, cardiac, hepatic, renal or other chronic pathologies) and year of notification.

Point 7: Describe any efforts to address potential sources of bias

Response 7: All samples sent by the different hospitals and laboratories have been included in the study. Possibly and especially in 2020 (due to the pandemic), 100% of the real cases of IPD will not be collected but we believe that the proportion of samples studied is representative thanks to randomness. As described in “Material and Methods”, some cases with unknown clinical outcome were excluded not to affect the real mortality and fatality rate as well as some strains with no available serotype to not change the proportion. We assume that missing serotype data are missing completely at random, that is the serotype distribution of serotyped cases is not biased. We assume that eliminated cases are randomly distributed.

Point 8: Results: are clearly presented. Please improve the quality of table

Response 8: We thank the reviewer for the suggestion. We have made the changes in the table. Please, see the new version submitted

Point 9: The discussion contains almost all the important information needed. However, the structure is a bit difficult to follow. Usually, the discussion is based on the following elements:

  1. Summary of the main results obtained 
  2. internal consistency and comparability of the data obtained
  3. followed by a general interpretation of the results in the context of other evidence
  4. limitations of the study performed
  5. implications of the results for practice, policy, and future research

In this paper even if most of these elements are there, they did not follow the appropriate structure and reader have to grasp into the text in order to found all of them. Please restructure the discussion. Moreover, in my view this study is highly important in terms of public health, but a very few elements and implications are discussed by authors. please add. this is an added value of your work.

Response 9: We thank the reviewer for the suggestion. Please, see the changes in the discussion.

Point 10: Conclusions are now missing. Please add 

Response 10: We thank the reviewer for the suggestion. We have made the changes in text

Reviewer 2 Report

Dear authors, congratulations on doing this piece of work. I think this is an important study, which shows the changes in epidemiology over time. It is certainly helpful to plan vaccines. I think this warrants publication but some minor changes are required

  1. What is the difference between fatality and mortality? I dont think there is a difference. Perhaps just use mortality.
  2. Can you make the discussion section less verbose, and perhaps break it down into smaller sections? Perhaps by age or serotype? for example line 153 to 243 is one single paragraph, and it is very dense

Author Response

Response to Reviewer 2 Comments

Point 1: What is the difference between fatality and mortality? I dont think there is a difference. Perhaps just use mortality.

Response 1: We thank the reviewer for the question and suggestion. “Case fatality rate” is defined as the proportion of events that are fatal within 28 days of onset. “Mortality rate” is the number of fatal events that occurs within 28 days per a population of 100 000 people (Reference https://doi.org/10.1161/01.STR.26.3.361). When we analyze mortality and serotype- associated case fatality rate, mortality rate is calculated considering the number of deaths by a serotype respect al IPD cases and fatality rate estimates the number of deaths by a serotype according the IPD cases by this serotype. In our study, we have measured the outcome case fatality rate and mortality rate based on these differences .

Point 2: Can you make the discussion section less verbose, and perhaps break it down into smaller sections? Perhaps by age or serotype? for example line 153 to 243 is one single paragraph, and it is very dense.

Response 2: We agree with the reviewer observation and we have followed the recommendation. See the new discussion.

Round 2

Reviewer 1 Report

satisfy with the changes